# HPV and Cervical Cancer Awareness and Screening Practices among Migrant Women: A Narrative Review

**DOI:** 10.3390/healthcare12070709

**Published:** 2024-03-23

**Authors:** Nuray Yasemin Ozturk, Syeda Zakia Hossain, Martin Mackey, Shukri Adam, Patrick Brennan

**Affiliations:** 1Faculty of Medicine & Health, The University of Sydney, Sydney, NSW 2000, Australia; nozt5374@uni.sydney.edu.au (N.Y.O.); martin.mackey@sydney.edu.au (M.M.); patrick.brennan@sydney.edu.au (P.B.); 2Faculty of Nursing, RAK Medical and Health Science University, Ras Al Khaimah 11172, United Arab Emirates; shukri@rakmhsu.ac.ae

**Keywords:** cervical cancer, cervical cancer screening, barriers, facilitators, migrant women

## Abstract

This narrative review explores the barriers and facilitators that migrant women face globally. The review explored a range of studies conducted in various countries, including the United States of America (USA), the United Kingdom (UK), Canada, Australia, and the United Arab Emirates (UAE). It also specialises in the experiences of migrant women living in Sydney, Australia, and women living in Ras Al Khaimah (RAK), UAE. Cervical cancer ranks as the fourth most prevalent form of cancer among women worldwide. It is the fourteenth most common cancer among women in Australia and the fourth most common cancer in the UAE. Despite the availability of vaccinations and cervical screening initiatives in many countries, including the USA, the UK, Canada, Australia, and the UAE, migrant women living in these countries continue to experience considerable health gaps when accessing cervical cancer screening services. Addressing these disparities is crucial to ensuring everyone has equal healthcare access. An electronic search was conducted using three databases to identify articles published between 2011 and 2021. Qualitative, quantitative, and mixed-methods research studies were included in the search. The identified factors were classified into categories of barriers and facilitators of cervical screening uptake, which were then sub-categorized. This narrative review examines the awareness of cervical cancer and screening behaviours, attitudes, barriers, and facilitators associated with cervical cancer screening. According to the study, several factors pose significant obstacles for migrant women worldwide, particularly those living in the USA, the UK, Canada, and Sydney, Australia, and Emirati and non-Emirati women (migrant women) residing in RAK when it comes to undergoing cervical cancer screening. These barriers include inadequate knowledge and emotional, cultural, religious, psychological, and organisational factors. On the other hand, social support, awareness campaigns, and the availability of screening services were found to promote the uptake of cervical cancer screening. The findings from this review suggest that healthcare providers should adopt culturally sensitive approaches to enhance awareness and encourage participation in screening programs among migrant women. Based on the findings of this narrative review, it is strongly suggested that healthcare providers and policymakers prioritise developing culturally sensitive screening initiatives for migrant women. It is essential to address the psychological and emotional barriers that prevent migrant women from accessing screening services. This can be accomplished by offering education and awareness campaigns in their native languages and implementing a community-based approach to encourage social support and increase awareness of cervical cancer and screening services. Furthermore, healthcare providers and organisations should provide educational tools that address common misconceptions based on cultural and religious factors that prevent women from accessing screening services.

## 1. Introduction

Cervical cancer is a type of cancer that occurs in the cells of the cervix, the lower part of the uterus that connects to the vagina. It is usually caused by the human papillomavirus (HPV), a sexually transmitted infection. Other risk factors for cervical cancer include smoking, a weakened immune system, long-term use of oral contraceptives, and a family history of cervical cancer. While cervical cancer is preventable with vaccination and early detection through regular screening, many women do not have access to these services or are unaware of their importance. Therefore, it remains a significant public health concern worldwide, with approximately 570,000 new cases and 311,000 deaths in 2018 [1]. However, the incidence of cervical cancer increased to 640,000, with 342,000 deaths globally in 2020 [1].

Cervical cancer ranks as the fourteenth most common cancer among women in the United States of America and Canada [2]. In the United Kingdom, cervical cancer is the 12th most common cancer among women overall and the second most common cancer among women aged 15 to 44 years. In Australia, cervical cancer is the fourteenth most common cancer among women, with an estimated 951 new cases and 256 deaths reported in 2019 [3]. Cervical cancer is more prevalent in the United Arab Emirates (UAE); it is the fourth most common cancer among women, with an incidence rate of 108 cases and a mortality rate of 56 per 100,000 women in 2018 [2].

Cervical cancer screening services have been instrumental in reducing morbidity and mortality rates globally [1]. Despite the availability of screening programs and strategies in countries such as the UK, Canada, the USA, and Australia, specific populations, such as migrant women, still face inequalities in accessing and utilising these services. Nonetheless, these countries have made significant efforts to promote equal access and ensure that screening services are available to all women, regardless of their ethnicity or socioeconomic status. Moreover, Australia and the United Arab Emirates (UAE) have significantly decreased cervical cancer incidence rates after implementing screening programs. In Australia, the National Cervical Screening Program (NCSP), introduced in 1991, has led to an 80% drop in cervical cancer incidence and mortality rates [3]. From 2000 to 2015, the overall incidence of cervical cancer in Australia decreased by 41.3%, and mortality rates decreased by 56.5% [3]. However, there is no comparable data regarding these statistics for the UAE. Given the adverse health impacts of cervical cancer, it is essential to explore the underlying factors that act as barriers and facilitators when accessing cervical screening behaviours among migrant women living in the US, the UK, Canada, Australia, and UAE. A comprehensive understanding of these factors is critical for developing effective strategies to enhance the health outcomes of migrant women across the globe. This will also assist policymakers and healthcare providers in delivering healthcare and education tools tailored to these population groups.

Globally, studies on cervical cancer have been primarily focused on high-income countries such as the United States, United Kingdom, Canada, and Australia and have neglected the experiences of low- and middle-income countries, including many Southeast Asian and Middle Eastern countries, where most migrant women originate from. Furthermore, there is limited research on cervical cancer among migrant women living in Sydney, Australia, and Emirati and non-Emirati (migrant) women in RAK, UAE, which can lead to inadequate preventative measures and treatment. This highlights the need for more targeted research and resources to increase awareness and improve access to screening and treatment for cervical cancer among migrant women in these regions. At the global level, there is a need to understand the attitudes and beliefs surrounding cervical cancer and screening among migrant women. To address this research gap, this narrative review of the literature will examine the attitudes and beliefs around cervical cancer and screening practices among migrant women coming from similar countries of origin and living in countries with high migrant populations, such as the USA, the UK, Canada, Australia, and the UAE. Additionally, it will explore the impact of various individual- and system-level factors that affect cervical cancer screening participation among migrant women, such as migration duration, language spoken at home, screening policies, and environmental and organisational factors.

Sydney’s migrant women were chosen because New South Wales (NSW) is one of the states in Australia with the highest proportion of the migrant population, with Western Australia having a migrant population of 35%, followed by Victoria (31%).

Cervical cancer is viewed differently in various cultures and societies, which can affect women’s willingness to seek screening services. Some cultures consider discussing gynaecological issues as taboo, which may make women feel ashamed or stigmatised [4,5]. Also, women’s societal roles and attitudes towards healthcare can vary, influencing their participation in screening programs. Therefore, it is crucial to comprehend these cultural disparities to create culturally sensitive services that cater to the population’s unique necessities. In the case of cervical cancer screening, cultural factors are significant barriers for migrant women worldwide [4,5,6,7]. This narrative review explores the cultural and societal barriers and facilitators for migrant women in the USA, the UK, Canada, Australia, and the UAE. Most importantly, we aim to examine the research that focuses on Muslim migrant women in Australia and Muslim women in the UAE and identify the factors that contribute to their cervical cancer screening knowledge and screening participation. It is essential to find out whether environmental factors and exposure to the healthcare systems contribute to women’s awareness and screening participation while living in Sydney, Australia, or RAK in the UAE.

One of the research gaps in this area is the limited understanding of the unique challenges women face in different population demographics. For instance, the cultural norms and beliefs of migrant women can differ significantly from the dominant culture in countries such as Australia, Canada, the UK, and the USA, making it challenging to engage them in cervical cancer screening initiatives. Similarly, women living in European or Middle Eastern countries, including women living in the UAE, may face barriers related to religious and cultural beliefs that prevent them from accessing cervical cancer screening services. Therefore, the study’s focus is on identifying barriers and facilitators for cervical cancer screening uptake among migrant women living in Sydney, Australia, and Emirati and non-Emirati women living in Ras Al Khaimah, UAE.

## 2. Materials and Methods

### Study Design

In early October 2021, an extensive search was conducted on PubMed’s Medline, PsychInfo, and CINAHL databases to locate original peer-reviewed articles published in English between 2011 and 2021. The search was all-encompassing, including quantitative, qualitative, and mixed-method studies.

To expand the search, full-text journal article references were manually examined for additional relevant articles. Table 1 lists the primary data sources used for this review.

The search was executed using the SPIDER strategy tool [8], incorporating specific keywords. Table 2 presents the keywords that were used. The Boolean operators “AND” and “OR” combined the keywords and refined the search results.

The study conducted an electronic search using three databases to identify articles published between 2011 and 2021. Qualitative, quantitative, and mixed-methods research studies were included in the search. The identified factors were then categorised into barriers and facilitators of cervical screening uptake, which were further sub-categorized. For this narrative review, we conducted an electronic search using three databases (PubMed, Scopus, and CINAHL) to identify articles published between 2011 and 2021. We included qualitative, quantitative, and mixed-methods research studies that explored cervical cancer awareness and screening practices among women in Australia, the United Arab Emirates, Canada, the United Kingdom, the United States, and New Zealand. We excluded studies that focused on men, studies that were not in English, and studies that were not published in peer-reviewed journals.

The screening and selection process was conducted systematically and transparently, following the Preferred Reporting Items for Systematic Reviews and Meta-Analyses (PRISMA) guidelines. All relevant studies retrieved from the database search were recorded and then digitised with their summaries. The studies were then examined by reading the full-text articles to see if they met the following inclusion criteria: (1) the study focused on women’s knowledge of cervical cancer and cervical cancer screening, (2) the study explored the barriers and facilitators of screening amongst migrant women, and (3) the studies were conducted in Australia, United Arab Emirates, Canada, the United Kingdom, the United States, or New Zealand. These countries were chosen as they have a high number of migrant populations, which makes them ideal subjects for study. However, certain research was deemed unsuitable for the following reasons: (1) the study only looked at the clinical aspects of cervical cancer; (2) it did not relate to cervical cancer screening practices; and (3) the research was in the form of literature reviews, case reports, editorials, or conference summaries.

The final number of studies included in the review was considered appropriate and sufficient to provide a broad understanding of cervical cancer screening practices among the target populations. In the selected articles, data were extracted through a comprehensive electronic search using three databases to identify studies published between 2011 and 2021. Qualitative, quantitative, and mixed-methods research studies were included in the search. The identified factors were classified into categories of barriers and facilitators of cervical screening uptake, which were then sub-categorized.

The findings were then synthesised by identifying common themes and patterns. This was carried out by summarising and organising the conclusions of each study and then comparing these findings to identify similarities and differences. The quality of the studies was assessed using established criteria. The criteria for selecting the articles used in our review were based on the studies’ relevance, reliability, and validity. We evaluated relevant articles that directly addressed our research objectives and ensured that our study was based on credible and up-to-date information. The reliability of the articles was determined by ensuring that the articles were subjected to peer review and published in reputable journals. The validity of the articles was determined by ensuring that they were based on reliable results, conclusions, and interpretations.

Furthermore, each article needed to serve a specific purpose in contributing to the overall objectives of our review. To guarantee that our findings were relevant and beneficial to the study populations, the context of each research was considered. We achieved our desired outcomes by using these established criteria for the article selection process.

After conducting an in-depth analysis and synthesis of various studies, we identified recurring themes and categories related to the barriers and facilitators of cervical screening uptake. These themes and categories were then organised into a comprehensive narrative review, which provided an overview of the factors that influence the uptake of cervical screening among migrant women in different countries, including Australia, New Zealand, the UK, the USA, Canada, and the UAE.

## 3. Results

The database yielded 186 articles, and 107 were excluded as they needed to meet the inclusion criteria based on their titles and abstracts. Following this, 79 articles were screened, and 54 were excluded since they did not match the inclusion criteria. A total of 25 articles met the inclusion criteria and subsequently were included in the full review. The reviewer also did a full-text screen and reference list scan. See Figure 1 for a search flow process chart.

This study reviewed 25 articles, consisting of 13 qualitative, 9 quantitative, and two mixed methods studies, that met the inclusion criteria. The gathered data from each publication were carefully analysed and tabulated in Table 3, which provides a comprehensive overview of the findings from the included studies. The data were analysed using a narrative review approach, identifying key themes and patterns across the studies and are presented in Table 3. Overall, the data analysis was conducted thoroughly and with attention to detail to ensure the accuracy of the findings. Through this literature review, we aimed to identify the primary barriers and facilitators of cervical cancer screening, as well as the individual- and system-based factors that impact the screening attendance and behaviour of migrant women. Our analysis of the existing literature suggests that most research on cervical cancer screening in migrant women has been carried out in the United States and the United Kingdom. However, there are limited studies that examine the experiences of migrant women in other countries, especially in Australia and the UAE. It is crucial to acknowledge that there exists a significant gap in the literature regarding the screening behaviours and attendance of migrant women. It is essential to recognise that various countries have their own distinct cultural and healthcare contexts, which can significantly influence the factors related to health and wellness. Therefore, it is essential to address this issue and find effective ways to bridge this gap, ensuring all women have access to quality healthcare and resources.

The review of the literature is presented under broad headings below (Table 3).

### 3.1. Individual-Level Factors

#### 3.1.1. Lack of Knowledge, Misunderstanding, and Education

Having accurate knowledge is crucial in making informed health decisions, particularly when it comes to cancer-related knowledge. This is notably relevant in the case of cervical cancer, where studies have consistently shown that individuals often lack a clear understanding of the disease’s risk factors and symptoms [5,6,7]. Therefore, due to a lack of knowledge and understanding, the number of individuals getting screened for cervical cancer has been decreasing, which can have a detrimental impact on their health and overall well-being. A recent quantitative study in the UAE revealed that 80% of surveyed women were unaware of a ‘precancerous cervical lesion’. Similarly, studies in Gulf countries have shown limited knowledge about cervical screening among women [7,9,10]. For instance, Jassim, Obeid, and Al Nashee’s (2018) [17] research found that 64% of Bahraini women had never heard of the term ‘pap smear test’. The findings of these studies emphasise the critical need to address the knowledge gap regarding cervical cancer in public health.

During the research, there were recurring instances of misconceptions related to risk factors for cervical cancer. Studies found that African migrants in the UK reported common misconceptions such as abortion, poor hygiene, and inserting fingers into the vagina as risk factors for cervical cancer [15,23]. Similarly, Uysal and Yildirim’s (2018) [28] study highlighted that migrant women in America reported that having multiple sexual partners or smoking were the most common risk factors for cervical cancer. Similarly, in another study conducted in America by Brown et al. (2011) [12], there was limited knowledge and confusion about the risk factors of cervical cancer and the HPV vaccination. Migrant women also reported experiencing ‘abnormal vaginal bleeding’ and ‘persistent abnormal discharge’ as symptoms of cervical cancer [15,29]. This was consistent with Ortashi et al.’s (2013) [7] study, where Arab migrant women believed that cervical cancer was only caused by a sexually transmitted infection (STI). In another study, African women thought that semen was a risk factor and that using condoms could prevent cervical cancer [11].

#### 3.1.2. Cultural Barriers: Cultural Shame and Stigma

Cultural shame and stigma often hinder cervical screening uptake amongst migrant women. In a qualitative study conducted in Canada by Vahabi and Lofters (2016) [29], undergoing a pap smear was depicted as ‘inappropriate sexual behaviour’ for Southeast Asian Muslim women. Like this study, Chang et al.’s (2013) [13] qualitative study on Chinese migrant women living in Australia reported that cancer and sexual health are sensitive topics within their communities. This same study also found that any concerns relating to female reproductive organs are not openly discussed between other migrant women or males or inter-generationally between older and younger women [13]. This is further supported by other studies reporting that the lower genital tract is sacred and part of the body only to be shared with husbands [10,29]. However, this was more common with studies based on Arab women.

### 3.2. Barriers and Facilitators

Lists of barriers and facilitators identified from studies under review are presented in Table 4 and are discussed below.

#### 3.2.1. Psychological Barriers

##### Embarrassment

Many women described temporary embarrassment and a lack of modesty when examined or treated during a cervical screening test [19,29]. The feelings of embarrassment among Asian migrant women arose from the idea that screening is ‘inconvenient’ due to exposure to specific body parts [13,19]. Furthermore, women from the Middle East and Southeast Asia reported embarrassment as a significant barrier when accessing cervical screening [23,29].

##### Fear

Middle Eastern and Asian migrant women stated that the fear of being diagnosed with cancer or worrying about the test results negatively impacted attending screening services [19,23]. Additionally, the fear of experiencing pain or discomfort during the cervical screening test was also a common barrier mentioned by migrant women [13,18]. In another study conducted in the US by Ndukwe, Williams, and Sheppard (2013), it was found that African migrant women reported feelings of fatalism and fear when asked about the HPV screening test.

#### 3.2.2. Practical Barriers

Practical barriers have been identified as a significant barrier to accessing screening services. Cervical cancer screening tests have been labelled as ‘traumatic’ and inconvenient by some African women living in Australia and Asian migrant women living in Canada [5,25]. The barriers to cervical cancer screening identified in a Canadian study include the lack of family physicians, inconvenient clinic hours, cultural barriers (e.g., modesty and language), and having a male physician [29]. Amongst the studies conducted in the UAE, Australia, and Canada, time off work, childcare services, and transport played a role in accessing cervical screening services [14,18,29]. Furthermore, the limited availability of pre-booked appointments and clinic hours made it difficult for many migrant women to participate in cervical screening tests [20,29]. In another study conducted in Canada, Chinese migrant women reported that the lack of time was also a common obstacle and was greatly affected by work and childcare commitments [30].

#### 3.2.3. Cognitive Barriers

##### Perceived Risks

Perceived risks of not having cervical cancer were another common barrier throughout the literature. For example, some women from the Middle East, Asia, and African ethnicities perceived they did not have cancer, which delayed screening attendance. A quantitative study in Oman supported these findings as Emirati women reported a low perceived risk of cervical cancer because they live a healthy lifestyle [10]. Being single and not sexually active was another primary reason European and Asian migrant women felt they were at low risk of being diagnosed with cervical cancer. Therefore, they did not attend screening practices [13,28].

##### Absence of Symptoms

The absence of symptoms during early cervical cancer diagnosis was a significant reason why screening behaviour was limited amongst some migrant women. However, for some migrant women from Asian backgrounds, the idea that the purpose of a pap smear test included looking for cancer cells in the absence of symptoms motivated them to attend screening services [19].

### 3.3. Healthcare Provider and System-Level Factors

#### 3.3.1. System-Level Factors

System-level factors include the barriers and facilitators of healthcare providers, such as logistical and organisational aspects. Healthcare provider experiences with a healthcare provider were both a barrier and a facilitator, according to some studies. One of the most reported barriers included the lack of availability of female healthcare providers and sub-optimal/poor/inadequate interpersonal, and communication skills [6,7,18,29]. In Vahabi and Lofter’s (2016) [29] study, migrant women from the Middle East and Asia reported that the main barrier to screening among service providers is the physician’s lack of sensitivity towards women’s needs for modesty. However, for some Asian migrants, female doctors from their culture were preferred because of modesty issues, and they could advise women about access to screening services. Several studies suggest that the physician’s recommendation for screening is considered the primary motivation for screening and is supported by many studies [6,10,19,25]. However, many women remembered previous negative screening experiences and male and female healthcare providers who had taken their samples, which affected their future screening attendance [11,12]. Many studies also report language as a primary barrier to cervical screening [11,19,29]. Migrant women have stated that language barriers make booking appointments difficult and affect communication between healthcare professionals [25]. Women in Asia and the Middle East reported that interpretation services were preferred and required for cervical screening [13,23].

Asian Muslim migrant women discussed the importance of having a female doctor perform their pap smear test as they felt embarrassed if examined by a male doctor [29]. In a study in England, female migrants from Poland, Slovakia, and Romania stated that they would only visit a clinic if the screening tests were conducted by a female doctor [16].

#### 3.3.2. Organisational Factors

According to research studies, the reminder letter sent out by the national screening program and a doctor’s recommendation are the most effective methods for promoting screening attendance among Middle Eastern and Asian women [19,29]. On the other hand, some of the significant barriers faced by Middle Eastern and Asian migrants in attending screening programs include a lack of childcare services as well as limited access to transportation [6,23]. However, the availability of free screening services has been identified as a significant motivator for migrant women residing in Australia. In a study conducted by Kwok, White, and Roydhouse (2011) [19], it was found that Chinese migrants in Australia greatly value the quality of healthcare available in the country, often appreciating it more than the healthcare systems in their home countries.

## 4. Discussion

Despite the increasing efforts to promote cervical cancer screening and raise awareness about its importance, there are still certain population groups that are not receiving the benefits of routine screening. This is a concerning issue, especially considering that cervical cancer can be fatal if not detected and treated early. Furthermore, migrant women around the world are disproportionately affected by cervical cancer, highlighting the need for greater attention and resources to be directed towards this population group. We must continue to work towards ensuring that all women, regardless of their background and circumstances, have access to effective cervical cancer screening programs. In many developing countries, access to preventative cancer care is limited, which poses a significant challenge for migrant women from these regions. Due to the lack of resources and awareness, these women often have little knowledge about the disease and are unfamiliar with the concept of cancer prevention. As a result, they may not receive timely screenings or take preventative measures that could help detect and treat cancer early on. For this reason, migrant women from Middle Eastern, South Asian, and Southeast Asian countries may be at a higher risk of late diagnosis of the disease. This underscores the importance of expanding access to cancer education and resources in developing countries to help promote the early detection and prevention of this life-threatening disease.

Regarding organisational factors, migrant women in Australia were motivated to undergo screening for various health issues due to the availability of free screening services. These services offered them access to vital healthcare facilities that they might not have been able to afford otherwise. The findings in this review highlight the importance of providing free screening services to underserved communities, mainly migrants who may face additional barriers to accessing healthcare services. Healthcare providers and screening programs should consider practical obstacles such as expenses, clinic atmosphere and facilities, clinic locations, and operating hours when delivering services to migrant women. By addressing these practical barriers with consideration to cultural, psychological, and financial barriers, more individuals can receive preventative care and reduce their risk of developing cervical cancer.

Furthermore, from the above review of the literature, it has been observed that migrant women, particularly those from conservative backgrounds, may feel uneasy when it comes to discussing matters related to sexual health. This is particularly prevalent among women from Middle Eastern and Southeast Asian regions. The reasons for this discomfort can be manifold, ranging from societal norms and cultural beliefs to a general lack of awareness or access to resources. Nonetheless, addressing these issues and providing safe and confidential spaces for these women to seek information and support is crucial. It is widely recognised that addressing matters concerning sexual health can be a sensitive subject, particularly within specific communities where cultural norms and values may pose significant barriers to open discussions on this topic.

Furthermore, it has been observed that females belonging to migrant communities often do not feel the need to discuss the human papillomavirus (HPV) due to the perception that it is solely a sexually transmitted disease. This assumption leads them to believe that they are not at risk of contracting HPV, which can prevent them from seeking necessary medical care or preventative measures. This indicates that ongoing support should be provided through healthcare services. Offering culturally competent care and service delivery whilst raising awareness and addressing common misconceptions about cervical cancer screening may enhance the experience for this population group. Health-related information should be linguistically appropriate, and the availability of interpreters should be considered. However, it is important to note that the availability of interpreters for discussing sensitive matters related to cervical cancer screening with migrant women is a debatable issue [18]. While interpreters can facilitate communication and improve understanding, some women may feel uncomfortable discussing such sensitive topics with a third party. In such cases, healthcare providers should offer alternative options such as language-appropriate educational materials or community-based approaches that encourage social support and increase awareness about cervical cancer and screening services. Ultimately, the goal should be to provide culturally sensitive and appropriate care to improve cervical cancer screening uptake among migrant women.

Moreover, many migrant women have expressed their concerns about attending screening services due to the fear of being diagnosed with cancer or the anxiety of waiting for the test results. This fear and worry can often act as a deterrent for women to attend regular cancer screenings, which can potentially lead to delayed diagnosis and treatment. This is a legitimate concern for women, and women must be encouraged to participate in screening services through culturally competent interventions.

Migrant women often encounter several obstacles when it comes to accessing cancer screening services. Among these, one of the most significant challenges is the fear of being diagnosed with cancer or the anxiety that arises while waiting for the test results. This fear can act as a significant barrier for women when it comes to attending regular cancer screenings, which can potentially lead to delayed diagnosis and treatment. The reasons behind this fear are manifold and may include language barriers, cultural differences, lack of awareness about available services, and mistrust of the healthcare system. All these factors can significantly impact a woman’s decision to seek cancer screening services and may exacerbate health disparities among migrant communities. This is a legitimate concern for women, especially those who are new migrants and already grappling with the challenges of settling in a new country. Therefore, it is crucial to recognise that women, mainly new migrants, may hesitate to attend screening services. Culturally competent interventions can help break down barriers and address their concerns, providing women with the support they need to attend these important screenings. By catering to their unique needs and circumstances, healthcare services can motivate women to take control of their health and ultimately lead happier, healthier lives.

Studies conducted in other regions and populations have also identified similar barriers and facilitators to cervical cancer screening uptake. For example, a study conducted in the United States found that cultural and linguistic barriers, lack of access to healthcare, and low health literacy were the main barriers to cervical cancer screening among Turkish migrant women [28]. Additionally, a study conducted in the United Kingdom found that fear, embarrassment, and lack of knowledge were significant barriers to cervical cancer screening [23]. In the UAE and RAK contexts, the identified barriers and facilitators are comparable to those in other regions. However, some unique factors may be contributing to the disparities in cervical cancer screening uptake. For example, in the UAE, cultural and religious beliefs may play a significant role in preventing women from accessing screening services. Women may also feel uncomfortable discussing sexual health issues with male healthcare providers, and this may act as a barrier to accessing screening services.

In Sydney, language barriers and a lack of familiarity with the healthcare system may prevent migrant women from accessing screening services. Furthermore, it is important to consider that the healthcare systems in different regions with varying cultural contexts, such as the UAE and Australia, may contribute to the disparities observed in cervical cancer screening uptake. Understanding these differences and tailoring interventions to address them could help improve access to screening and reduce health disparities. For instance, in the UAE, the healthcare system is mainly centralised, and access to screening services may be limited in rural areas. In Australia, access to the healthcare system may be limited for migrant women due to unfamiliarity with the system and difficulty navigating healthcare services. In conclusion, while the identified barriers and facilitators to cervical cancer screening uptake are comparable to those found in other regions and populations, unique cultural, linguistic, and healthcare system factors may contribute to the disparities in the UAE and Australia. Addressing these unique factors through culturally sensitive approaches, community-based education, awareness campaigns, and improving access to screening services can help reduce disparities and improve cervical cancer screening uptake among migrant and non-migrant women in these contexts.

Moreover, it is important to note that cervical cancer should not be viewed in isolation but as part of a holistic prevention strategy approach. This approach should incorporate other preventive activities such as breast cancer screening, biochemical testing, and blood pressure monitoring. These interventions will not only help prevent cervical cancer but will also have an added value towards general health prevention.

The study’s findings strongly suggest that healthcare providers and policymakers should prioritise the development of culturally sensitive screening initiatives for migrant women. To ensure that migrant women can access screening services, it is crucial to address the psychological and emotional barriers that prevent them from seeking help. One way to accomplish this is by providing education and raising awareness about cervical cancer and screening services in the native languages of these women. A community-based approach can also be implemented to encourage social support and increase awareness of the importance of screening. In addition, healthcare providers and organisations must provide educational tools that address common misconceptions based on cultural and religious factors that prevent women from accessing screening services. It is crucial to prioritise developing culturally sensitive screening initiatives for migrant women and address the psychological and emotional barriers that prevent them from accessing screening services. By adopting culturally sensitive approaches and implementing a holistic prevention strategy, we can ensure that everyone will have equal healthcare access and reduce the incidence of cervical cancer globally. Healthcare providers and policymakers must invest in education and awareness campaigns in their native languages and implement a community-based approach to encourage social support and increase awareness of cervical cancer and screening services. By adopting culturally sensitive approaches and implementing a holistic prevention strategy, we can ensure that everyone will have equal healthcare access and reduce the incidence of cervical cancer globally.

Although few studies explore cervical cancer screening amongst migrant women living in Sydney or RAK, there is very little research that examines the cervical cancer screening needs of both migrant communities. It is crucial to conduct further research about cervical cancer screening to increase awareness and promote programs that focus on raising screening participation rates within communities. This will help educate people about the importance of early detection and encourage them to prioritise regular screenings.

Overall, there is a lack of research on cervical cancer awareness and screening practices among migrant women globally. However, it is also essential to acknowledge that this review, being a narrative literature review, has its limitations. A narrative review differs from a systematic review because it does not have to adhere to a particular research question, methodology, or an extensive search strategy [8]. Hence, despite our rigorous efforts to incorporate the most pertinent research, some articles may have been inadvertently overlooked during our search across various databases.

Moreover, upon focusing the online search on Australia and the UAE, it became evident that there are notable gaps in the existing literature. Most studies reviewed in this analysis originated from the UK and Canada. While the obstacles encountered by migrant women were still identified, it is crucial to acknowledge that these barriers may not be identical to those encountered by women residing in other countries. Thus, it is essential to undertake research specific to these regions to better understand the challenges faced by various population groups.

While studies have been conducted in many countries, including the USA, the UK, and Canada, there are still gaps in the current literature that need to be addressed. Moreover, it is important to recognise that the experiences of women can vary greatly depending on their location and cultural context. For instance, women residing in Australia and the UAE may face unique challenges that are specific to their respective countries. Therefore, there is a need for research that is specific to these regions to better understand the challenges faced by women and develop effective solutions. While reviewing multiple studies, it was observed that many did not mention the time the migrants had lived in the host country. Additionally, data were collected primarily in English, indicating that some participants were proficient in the language and therefore had been residing in the country for several years. The lack of discussion surrounding the duration of residency restricted the examination of acculturation among the women in these studies, thereby underlining the need for more comprehensive research that considers all such factors. Furthermore, it is worth noting that there is a shortage of research studies conducted on Middle Eastern women compared to their Asian and African counterparts who have migrated to different parts of the world. This highlights the need for further investigations that explore the diverse ethnic backgrounds of women from this region. Furthermore, while discussing the role of religiosity in the lives of migrant women, the current literature focuses solely on Muslim migrant women without considering the experiences of women belonging to other religions. This limitation also reinforces the need for more inclusive and comprehensive research studies.

## 5. Conclusions

This narrative review extensively analyses the factors that affect cervical cancer screening uptake amongst migrant women. Our analysis highlights the need for targeted health promotion and educational programs to address sociocultural attitudes towards women’s sexual health in migrant communities.

The results of this narrative review could have a considerable impact on healthcare policies, specifically in the effort to address the disparities in cervical cancer screening amongst migrant women across the globe.

Although this review provides valuable insights into the barriers encountered by migrant women in various countries, such as Australia, the United Kingdom, the United States, Canada, and the UAE, there are still some gaps in the literature that need to be addressed. This will help provide a more comprehensive understanding of the issues and challenges faced by migrant women worldwide. Further research is necessary to understand the levels of knowledge and awareness among migrant women to reduce the morbidity and mortality rates caused by this disease. This will help to inform public health policies and enhance our comprehension of the issue. Healthcare providers and policymakers should focus on creating culturally appropriate interventions that address the psychological, emotional, cultural, and religious barriers that prevent migrant women from accessing screening services. Education and awareness campaigns in native languages and a community-based approach can encourage social support and therefore will also increase awareness and screening attendance. Such effective strategies must be implemented to ensure easy access to healthcare and services for eligible migrant women. Overall, by addressing the barriers and facilitators that influence screening uptake, the incidence and mortality rates of cervical cancer can be significantly reduced among these population groups.

The findings show that despite the availability of screening services, a considerable number of women in these regions are not utilising them, which may lead to delayed diagnosis and unfavourable outcomes. The issue of the underutilisation of screening services is not limited to specific regions or countries, but it is a global phenomenon. Despite the presence of well-established healthcare systems and screening services in many countries, a significant number of women are still not accessing them. Furthermore, the review identified the challenges that impede cervical cancer screening in these areas, such as language barriers, lack of awareness, and cultural beliefs, which have policy implications. The potential strategy may include the provision of education and awareness campaigns to inform women about the benefits of screening and collaboration with community organisations to address cultural barriers. By implementing these strategies, we can help to promote the early detection and treatment of cervical cancer, leading to better health outcomes for individuals and communities around the world.

In conclusion, the findings of this narrative review highlight the need for policymakers and healthcare providers to develop culturally sensitive and context-specific strategies to address the disparities in cervical cancer screening among migrant women in different healthcare systems. By implementing such strategies, healthcare systems in the USA, the UK, Canada, Australia, and UAE can ensure that all eligible women have equal access to screening services, thereby reducing cervical cancer morbidity and mortality rates, especially among migrant women.

## Figures and Tables

**Figure 1 healthcare-12-00709-f001:**
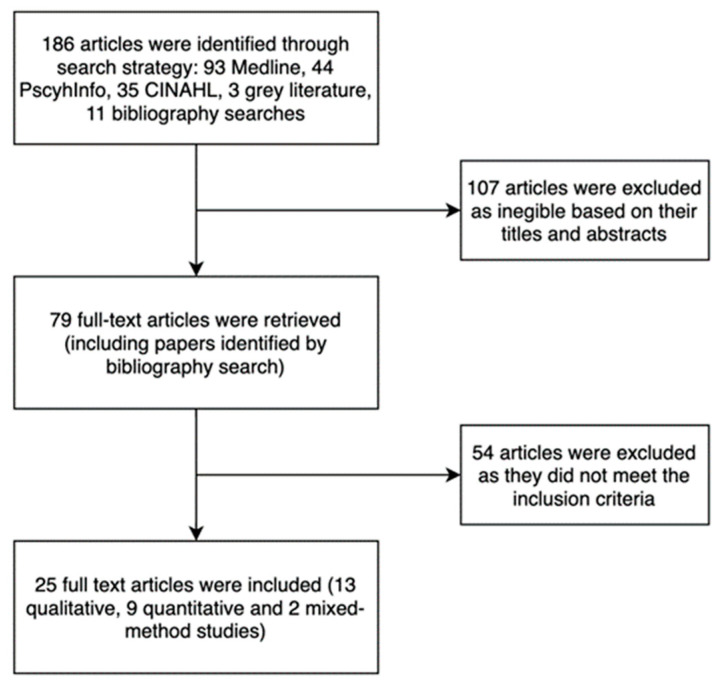
Search flow process chart.

**Table 1 healthcare-12-00709-t001:** Data sources.

Data Sources	
Databases used	Medline, Pyschinfo, CINAHL
Other electronic sources	Google Scholar
Hand searches	Reference lists of all included articles were hand-searched
Grey literature	

**Table 2 healthcare-12-00709-t002:** Search strategy and keywords used.

SPIDER	Search Term
S—Sample	“migrant* women” OR “immigrant* women” OR “migrant*”
PI—Phenomenon of Interest	“Cervical screening” OR “pap smear” OR “cervical screening test” or “CST” or “cancer screening” OR “pap test” OR “human papillomavirus” or “HPV”
D—Design	“interview” OR “questionnaire” OR survey” OR “focus group discussion” OR “FGD” OR “observation”
E—Evaluation	“barrier” OR “facilitator” OR “motivator” OR “attitude*” OR “belief*” OR “knowledge” OR “awareness”
R—Research Type	“Qualitative” OR “quantitative” OR “mixed method”

**Table 3 healthcare-12-00709-t003:** General Attributions and Findings of Included Articles.

Author (Year)	Country	Sample	Study Design	Types of Outcomes	Main Findings
Al-Hammadi et al., 2017 [9]	United Arab Emirates (UAE)	N = 599	Quantitative (cross-sectional survey).	Knowledge, attitudes, and practice towards and reasons for undergoing the pap test in the UAE.	Knowledge about the pap smear test was limited, and awareness that they should undergo the pap smear test every three years even with an initial negative/normal pap smear result was abysmal. Despite the positive attitude of the women towards the pap smear test, almost 80% of the women surveyed had no knowledge of precancerous lesions.
Alissa, 2021 [6]	Saudi Arabia	N = 467	Quantitative (online survey).	To investigate the knowledge of pap testing and screening attendance amongst Saudi Arabian Women.	This study examined the knowledge of pap smear tests and the intention to undergo pap testing. It found that study participants have above-average levels of knowledge and the intention to undergo pap smear testing. However, these results were not consistent with adopting preventive behaviours. The study revealed a significant correlation between demographic factors (age and marital status) and the intention to attend screening services.
Alwahaibi et al., 2017 [10]	Oman	N = 494	Quantitative (surveys).	Knowledge, attitude, and practice of pap smear among Omani women.	Participants included a comparison of patients, staff, and students. The knowledge of pap smear among patients, staff, and students was as follows: 56.9%, 56.4%, and 23.6% The common barrier that prevents the uptake of pap smear among the three groups was their belief that they have a healthy lifestyle.
Anaman-Torgbor, King, and Correa-Velez, 2017 [11]	Australia	N = 19 (10 refugee and 9 non-refugee) African	Qualitative (interviews).	Barriers and facilitators of cervical cancer screening practices.	The results did not show major differences between refugee and non-refugee women. The following barriers were reported: lack of knowledge about cervical cancer and pap smears, absence of symptoms, embarrassment, fear, gender of the doctor, lack of privacy, cultural and religious beliefs, and healthcare system factors.
Brown et al., 2011 [12]	America	N = unknown Haitian, African, English-Speaking Caribbean, and African American	Qualitative (six × focus groups).	To describe cervical cancer screening knowledge, attitudes, beliefs, and practices amongst ethnically diverse Black women.	This study found that there was limited knowledge and confusion amongst the ethnic groups and their view on cervical cancer and its risk factors, the pap test, and the human papilloma virus (HPV) along with its association with cervical cancer. However, there were also some differences between ethnic groups in knowledge, cultural beliefs, and practices about cervical cancer. The main motivator to seek screening services was a positive patient–doctor relationship. Barriers included a busy work schedule, fear of the unknown, lack of insurance, unemployment, cost, and fear of disclosing immigration status. Suggested interventions include culturally based strategies through current social networks and cervical cancer education.
Chang et al., 2010 [13]	Australia	N = 171 (78 Caucasian and 93 Chinese)	Qualitative (questionnaire).	Cervical cancer screening beliefs and practices of Chinese and Caucasian mother–daughter pairs.	For migrant women who had attended one pap test in their life, there were no reported ethnic differences in the group of migrant women who had recommended attending screening often. Lower acculturation rates were associated with higher screening attendance rates between both mothers and daughters.
Cullerton et al., 2016 [14]	Australia	Seven CALD groups (Arabic-speaking, Bosnian, South Asian (including Indian and Bhutanese), Samoan and Pacific Island, Spanish-speaking, Sudanese, and Vietnamese)	Qualitative (pre- and post-education session questionnaires).	Cancer screening education and if it can change knowledge and attitudes among culturally and linguistically diverse communities.	After the provided education sessions, the knowledge of participants increased, and some attitudes towards participation became much more positive. Participants intended to participate in future screening
Ekechi et al., 2014 [15]	UK	N = 937 Age = 18–78 Black women (predominantly from African or Caribbean backgrounds)	Quantitative (questionnaires).	Knowledge of cervical cancer and attendance of screening practices.	Women that had higher educational qualifications (*p* < 0.001) and that were born in the UK (*p* = 0.11) had a better understanding and increased knowledge about cervical cancer risk factors. Older age was associated with increased knowledge of the symptoms (*p* < 0.001). The main barriers that affected screening were fear of the procedure of the test (18%) and low perceived risk (18%) of cervical cancer.
Jackowska, et al., 2012 [16]	UK	N = 31	Mixed qualitative methods. (1) Interviews with Central Eastern European migrants (*n* = 11), (2) focus group including three Polish, one Slovak, and 1 Romanian women, (3) interview of Polish (*n* = 11), Slovak (*n* = 7), and Romanian (*n* = 2) women.	The attitudes and beliefs of Polish, Slovak, and Romanian women towards cervical cancer screening.	Awareness of the cervical screening programme was good, but an understanding of the purpose of attending screening services was sometimes limited. Women appreciated that the screening is free and that reminders are sent. However, some were concerned about the age of the first invitation.
Jassim, Obeid, and Al Nasheet (2018) [17]	Bahrain	N = 45	Qualitative (interviews).	Knowledge, attitudes, and practices regarding cervical cancer and screening among women visiting primary healthcare centres in Bahrain.	Over 64% (194 participants) had never heard of a pap smear procedure, and only 3.7% (11 participants) had heard about the human papillomavirus (HPV) vaccine. Most participants felt embarrassed when examined by a male doctor (250, 83.3%), and few underwent a pap smear screening if they were never married (69, 23.0%).
Khan and Woolhead, 2015 [18]	Dubai (UAE)	N = 13 (Six Southeast Asian women, seven Emirati women)	Qualitative (13 in-depth interviews).	Perspectives on cervical cancer screening among educated Muslim women in Dubai	Four themes emerged: (1) CC was considered a ‘silent disease’, also associated with extramarital sexual relations which negatively influenced screening uptake, (2) fear, pain, and embarrassment, (3) growing mistrust of allopathic medicine, and (4) became aware of screening when pregnant or seeking fertility.
Kwok, White, and Roydhouse, 2011 [19]	Australia	N = 18 Chinese background	Qualitative (in-depth interviews in their native languages).	Differences in barriers and facilitators of cervical cancer screening within Chinese Australian migrant women.	Knowledge of cervical cancer awareness and practices were low, and only a few of the participants understood the advantages and importance of screening. The recommendation to seek screening services from a doctor was a strong motivator of screening, and returning to attend screenings was encouraged by having a female Chinese doctor to conduct the cervical screening test. Receiving a reminder letter and the screening service being free of cost were also motivators for undergoing screening.
Lofters et al., 2011 [20]	Canada	N = 455,864 (East Asia Pacific (*n* = 128,965), Eastern European + Central Asia (*n* = 67,845), Latin American + Caribbean (*n* = 70,184), Middle Eastern + North African (*n* = 33,649), South Asia (*n* = 88,107), Sub Saharan African (*n* = 26,125), USA, AUS + NZ (*n* = 10,003), and West European (*n* = 30,167)	Quantitative (data used from Canadian National Population Health Survey (1998–1999).	The predictors of low cervical cancer screening through sociodemographic, the current healthcare system, and migration, varying from different ethnic backgrounds.	This study found that the country of origin did not affect a lack of cervical cancer screening. The main barriers to screening included living in low-income neighbourhoods, not having a primary care patient enrolment model, not having access to female providers, or not having access to providers from the same region. For all the women, the most common barrier was not having a female health provider, with values ranging from 16.8% [95% CI 14.6–19.1%] for women from the Middle East and North Africa to 27.4% [95% CI 26.2–28.6%] for women from East Asia and the Pacific.
Lofters et al., 2017 [21]	Canada	N = 30 women (recruited over a 3-month period) Convenience sampling Ages—21–69 Participants were self-identified as Muslim, foreign-born, and had good knowledge of English.	Qualitative (self-completed questionnaire).	The acceptability of HPV self-sampling amongst Muslim migrant women and the barriers to seeking cervical cancer services including self-sampling.	This study demonstrated that more than half of the participants indicated that pap smear tests can cause cervical infection, and 46.7% of participants found that the pap smear test is an invasion of privacy. Self-sampling was discussed for this study, and most migrant women agreed that they would prefer this more than a health service provider conducting the pap smear test. The main barriers to self-sampling in this study included the perceived cost and lack of confidence in doing the test. The facilitators included convenience and the preservation of privacy.
Madhivanan et al., 2016 [22]	USA	N = 35 Hispanic/Latino women	Qualitative (six × focus group discussions).	Barriers and facilitators of cervical cancer screening practices.	The main facilitator of cervical cancer screening discussed in this study included family support (especially from female relatives). Participants also reported prioritising family health more than their own and stated that they avoided screening due to fatalistic beliefs about cancer. Other barriers included a fear that a pap test might remove the uterus, discomfort about a male doctor conducting the screening, and concerns of testing stigmatising the participants as being sexually promiscuous or having an STD. Future implication suggested targeting women of all ages as younger females usually turn to older female relatives to seek advice.
Marlow, Waller, and Wardle, 2015 [23]	UK	N = 54 (Indian, Pakistani, Bangladeshi, Caribbean, African, Black British, Black other and White other (*n* = 43), and White British women (*n* = 11))	Qualitative (interviews).	Self-perceived barriers to cervical cancer screening services within ethnic minority groups in comparison to White British women.	Migrant women felt as if there was a lack of awareness and education about cervical cancer in their community. Many women did not recognise the terms ‘cervical screening’ or ‘smear test’. Fifteen of the forty-three migrant women had delayed screening or had never participated in screening. The main barriers to cervical cancer screening that were raised by all the migrant women were emotional (embarrassment, fear, shame), cognitive (low perceived risk, absence of symptoms), and practical (lack of time). Asian women reported emotional barriers the most when compared to other migrant women from the study. Low perceived risk of cervical cancer was associated with the beliefs about having sex outside of marriage, and some women felt that a cervical cancer diagnosis is considered shameful. All of the women remembered negative experiences, and this has been a suggested factor that acted as a barrier to screening.
Marlow, Wardle, and Waller, 2015 [4]	UK	N = 720 Ages: 30–60 years Ethnic backgrounds: Indian, Pakistani, Bangladesh, Caribbean, African, and White British	Qualitative (structured interviews).	Attitudes of cervical screening and non-attendance amongst BAME (Black, Asian, and minority ethnic) women.	Migrant women were less likely to attend screening services when compared to British women (44–71% vs. 12%). Two groups amongst migrant women were identified—the disengaged and the overdue. Not being able to speak English and low education levels were associated with being disengaged, and older age was associated with being overdue. Three main barriers were the low perceived risk of being diagnosed with cervical cancer due to sexual inactivity, the lack of necessity to screen without symptoms, and the difficulty finding an appointment that fits with other commitments.
Ndukwe, Williams, and Sheppard, 2013 [24]	US	N = 38 Ages: 20–70 Ethnic backgrounds: Ghana, Nigeria, Cameroon, Zambia, and Ivory Coast	Qualitative (focus groups/interviews).	Knowledge, awareness, and perspective of both cervical and breast cancer.	The awareness of cervical cancer risk factors and symptoms were low. Barriers to both breast and cervical cancer included barriers to accessing services, feelings of fatalism, stigma, privacy and confidentiality concerns, and fear. Factors that motivated screening were cancer death in the family, experiencing cancer symptoms, and reminders from primary care providers.
Ogunsiji et al., 2013 [5]	Australia	N = 21 Snowball sampling used Ethnic backgrounds: West-African migrant women	Qualitative (in-depth interviews). 90 min each.	Knowledge, awareness, usage, and attitudes towards cervical cancer of West African migrant women living in Australia.	The following three themes were found through this study: (1) knowledge of cancer screening, (2) attitudes and beliefs towards cancer screening, and (3) and the utilisation of cancer screening. Despite where in Africa the migrants were born in, 20 of the 21 participants had no knowledge of cancer screening before they migrated. Most participants also had a negative attitude towards screening. Women who gave birth after their migration to Australia were more likely to seek cervical cancer screening services. Older women who had passed their child-bearing years or that did not regularly visit healthcare services were more likely to have limited knowledge and awareness of cervical cancer screening services.
Ortashi et al., 2013 [7]	UAE	N = 640 Women aged 18–50 years	Quantitative (cross-sectional survey).	Awareness and knowledge about human papillomavirus infection and vaccination.	Only 29% of participants heard of the HPV infection, and 15.3% recognised it as an STI. Approximately 22% women knew what the HPV vaccine was, and 28% recognised the vaccine as a preventative measure against cervical cancer.
Redwood-Campbell et al., 2011 [25]	Canada	N = 11 Participants were newly immigrated (1–5 years) Ages: 35–69 Married Language groups: Arabic, Cantonese, Somali, Dari (Afghanistan), and Spanish (Latin America)	Qualitative (two focus group interviews for each group, one in English and one in the native language).	The differences and similarities among multiple groups of migrant women and Canadian-born women of low socioeconomic status + the barriers and facilitators that are associated with cervical cancer screening to inform and direct suitable strategies to help raise awareness amongst these under-screened groups.	The participants all displayed a strong need for information on the importance of cervical cancer screening and how it is performed by health professionals. The participants stated that videos and group discussions are the preferred methods of increasing their awareness. Women felt proactive about seeking cancer screening services and prevention methods despite this not being the norm in their home countries. Only Chinese and Arabic migrant women discussed modesty and embarrassment as barriers to screening. A female doctor was preferred more than language congruence between the provider and the patient. This study concluded that knowledge gaps need to be addressed and personal approaches need to be used to increase migrant women’s knowledge and awareness. Invitations to screening have also been suggested to reduce feelings of stigma and fear amongst migrant women from lower socioeconomic status and different ethnic groups.
Robison et al., 2014 [26]	UK	N = 228 (Chinese *n* = 96 and Southeast Asian *n* = 132)	Quantitative (cross-sectional survey). Categorical variables were compared by Fisher’s exact test. Mean scores of correct answers on the knowledge questions were compared through *t*-tests and analysis of variance.	Knowledge awareness and prevention strategies of cervical cancer screening among Chinese American women compared to Southeast Asian American women.	Chinese women had higher levels of college education (67%) when compared to Southeast Asian women (37%) (*p* < 0.0001). Among both migrant groups, 25% of the participants had never attended a pap test or were unsure if they had ever attended a pap test. Chinese migrants showed a greater lack of knowledge about the relationship between HPV and cervical cancer (mean 2.9 out of 8 questions) in comparison to Southeast Asian women (mean 3.6 out of 8 questions, *p* = 0.02). Despite ethnic subgroups, education, and income levels, all participants had poor knowledge and awareness of HPV and cervical cancer screening.
Team, Manderson, and Markovic, 2013 [27]	Australia	N = unknown European	Qualitative (in-depth interviews).	Women’s health-related behaviours affecting the participation in breast and cervical cancer screening.	Participants from this study had grown up in the former Soviet Union where health checks were compulsory and where timing and frequency of appointments was the responsibility of the doctors. After migrating to Australia, women continued to believe that appointments and check-ups were still the responsibility of the doctors, which motivated them to maintain this. Women argued that sexual health screening was important to them and that health professionals should take the lead role to guarantee that every female can participate.
Uysal and Yildirim, 2018 [28]	USA	N = 156 Turkish migrant women Ages: 35–45	Quantitative (questionnaire-based survey) SPSS was used to compute frequency and descriptive statistics.	Knowledge and awareness of pap smear testing and the risk factors of cervical cancer among female Turkish migrants.	This study found that more than half of the Turkish migrant women (66%) reported attending a cervical screening test at least once in their lives. Over two-thirds (85.8%) of participants knew that abnormal vaginal bleeding, vaginal infections (78.2%), sexual activity with a man who has had multiple partners (61.5%) and having multiple sexual partners (61.5%) increases the risks of cervical cancer. Through the results of the multivariate regression analysis, it was determined that the age of Turkish migrant women (OR 11.3, 95% CI 5.1–25.2, *p*: 0.000) and the number of children (OR 3.4, 95% CI 1.7–6.9, *p*: 0.000) are factors that affect pap smear testing attendance. Furthermore, it was found that low levels of knowledge about cervical cancer impacted accessing cervical cancer screening services.
Vahabi and Lofters, 2016 [29]	Canada	N = 30 Convenience sample Ages: 21–69 Characteristics of participants: foreign-born, Muslim, and good knowledge of English	Mixed-methods study design (focus groups).	To explore Muslim migrant women’s views on cervical cancer screening and HPV self-sampling.	The barriers for cervical cancer screening identified in this study included lack of family physician, inconvenient clinic hours, cultural barriers (e.g., modest and language), and having a male physician. The results also showed that HPV self-sampling is preferred to the traditional health service provider administering the pap smear test.
Xiong et al., 2010 [30]	Canada	N = 64,604 Asian immigrants	Quantitative (data from the Canadian Community Health Survey Cycle, 2003). (Multivariate logistic regression analyses were conducted to compare rates and determinants of cervical cancer screening between both Asian and non-Asian immigrants).	Barriers associated with cervical cancer screening amongst Asian Canadian immigrant and non-migrant women.	Asian immigrants had drastically lower rates of cervical cancer screening (52%) when compared to non-migrants (72%). The main barriers mentioned throughout this study were lack of necessity and time.

Note. Of the 25 studies reviewed, there were 13 qualitative studies, 9 quantitative studies and two mixed-method studies.

**Table 4 healthcare-12-00709-t004:** Barriers and facilitators towards Cervical Cancer Screening among migrant women identified throughout the literature.

Barrier or Facilitator	Author
Inadequate knowledge of cervical cancer screening services	Anaman-Torgbor, King, and Correa-Velez, 2017 [11]; Brown et al., 2011 [12]; Chang et al., 2010 [13]; Cullerton et al., 2016 [14]; Ekechi et al., 2014 [15]; Kwok, White, and Roydhouse, 2011 [19]; Marlow, Wardle, and Waller, 2015 [4]; Ogunsiji et al., 2013 [5]; Uysal and Yildirim, 2018 [28]
Adequate knowledge yet still low participation rates	Jackowska et al., 2012 [16]
Healthcare providers were considered responsible to facilitate screening	Team, Manderson, and Markovic, 2013 [27]
Socio-demographic factors including age, marital status, and education were barriers	Marlow, Wardle, and Waller, 2015 [4]; Redwood-Campbell et al., 2011 [25]
Socio-demographic such as age, marriage and education status	Ekechi et al., 2014 [15]
Fear of pain	Anaman-Torgbor, King, and Correa-Velez, 2017 [11]; Ekechi et al., 2014 [15]; Vahabi and Lofters, 2016 [29]; Marlow, Wardle, and Waller, 2015 [4]; Brown et al., 2011 [12]
Fear of diagnosis as a barrier	Brown et al., 2011 [12]; Marlow, Wardle, and Waller, 2015 [4]; Redwood-Campbell et al., 2011 [25]
Fear of diagnosis as a facilitator	Marlow, Wardle, and Waller, 2015 [4]
Felt a lack of modesty in regard to screening services	Marlow, Wardle, and Waller, 2015 [4]; Redwood-Campbell et al., 2011 [25]; Vahabi and Lofters, 2016 [29]
Cultural beliefs acted as a barrier	Anaman-Torgbor, King, and Correa-Velez, 2017 [11]; Redwood-Campbell et al., 2011 [25]
Embarrassment regarding cervical cancer screening activities	Anaman-Torgbor, King, and Correa-Velez, 2017 [11]; Marlow, Wardle, and Waller, 2015 [4]; Redwood-Campbell et al., 2011 [25]; Vahabi and Lofters, 2016 [29]
Embarrassment through stigma of cervical cancer misconceptions of having an STD or many sexual partners	Marlow, Wardle, and Waller, 2015 [4]; Redwood-Campbell et al., 2011 [25]; Vahabi and Lofters, 2016 [29]
Confidentiality	Nduwke, Williams, and Sheppard, 2013 [24]; Redwood-Campbell et al., 2011 [25]; Vahabi and Lofters, 2016 [29]
Women expressed religion to be a barrier from the healthcare providers’ perspective	Vahabi and Lofters, 2016 [29]
Cost and lack of health insurance to cover cervical cancer screening as a barrier	Brown et al., 2011 [12]
Living in low-income neighbourhoods as a barrier	Lofters et al., 2011 [20]
Appreciation for the Australian healthcare system due to free screening services	Kwok, White, and Roydhouse, 2011 [19]
Appreciation of free screening in other countries (UK)	Jackowska et al., 2012 [16]
Low perceived risk due to thoughts of not being affected by cervical cancer	Chang et al., 2010 [13]; Marlow, Wardle, and Waller, 2015 [4]
Low perceived risk because not married, sexually active, or do not have an STD	Chang et al., 2010 [13]; Madhivanan et al., 2016 [22]; Marlow, Wardle, and Waller, 2015 [4]
Absence of symptoms and therefore unnecessary	Marlow, Wardle, and Waller, 2015 [4]; Xiong et al., 2010 [30]
Discomfort of male doctor conducting screening test was a barrier	Lofters et al., 2011 [20]; Madhivanan et al., 2016 [22]; Redwood-Campbell et al., 2011 [25]; Vahabi and Lofters, 2016 [29]
Patient–doctor relationship as the main facilitator to attend screening	Brown et al., 2011 [12]
Recommendations from a doctor as the main facilitator	Kwok, White, and Roydhouse, 2011 [19]
Returning to screening due to doctor being the same ethnicity as the migrant	Kwok, White, and Roydhouse, 2011 [19]
Female doctor conducting pap smear test as a facilitator	Redwood-Campbell et al., 2011 [25]; Vahabi and Lofters, 2016 [29]; Kwok, White, and Roydhouse, 2011 [19]
Follow-up reminders/letters sent for routine screening as a facilitator	Jackowska et al., 2012 [16]; Kwok, White, and Roydhouse, 2011 [19]
Busy work schedule/lack of time	Brown et al., 2011 [12]; Ogunsiji et al., 2013 [5]; Vahabi and Lofters, 2016 [29]; Xiong et al., 2010 [30]; Team, Manderson, and Markovic, 2013 [27]
Inconvenient clinic hours	Vahabi and Lofters, 2016 [29]
Language as a barrier	Cullerton et al., 2016 [14]; Redwood-Campbell et al., 2011 [25]; Vahabi and Lofters, 2016 [29]
Misconception of pap test causing cervical cancer	Lofters et al., 2017 [21]
Misconception of sperm causing cervical cancer	Ekechi et al., 2014 [15]
Misconception of pap smear removing the uterus	Madhivanan et al., 2016 [22]
Family support as a facilitator	Madhivanan et al., 2016 [22]
Death in the family caused by cancer as a facilitator	Nduwke, Williams, and Sheppard, 2013 [24]
Passed child-bearing age and therefore considered screening not importance	Ogunsiji et al., 2013 [5]
Abortion, lack of hygiene, and the insertion of fingers into the vagina were risk factors of cervical cancer	Ogunsiji et al., 2013 [5]

## Data Availability

All data generated or analysed during this study are included in this published article.

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
