# Peer review of "HPV and Cervical Cancer Awareness and Screening Practices among Migrant Women: A Narrative Review"

_healthcare, 2024, doi:10.3390/healthcare12070709_

Round 1
Reviewer 1 Report (Previous Reviewer 1)
Comments and Suggestions for Authors
The paper has been well-revised but again needs to be more concise and focused on its topic. It remains unclear from the manuscript why two sites in the title, Sydney and Ras Al-Khaimah, which are 12,000 km apart and lack any cultural, geographical, or ethnic similarities, were chosen. A detailed explanation of the rationale behind selecting these diverse locations would enhance the analysis. While this choice might be understandable for primary research, for a review, it could lead to scattered findings due to differences in socio-cultural, medical, economic, and geopolitical contexts.
The study includes numerous studies from the UK, the Middle East (other than UAE), Canada, and the US. Please amend the title to reflect what has been done and what is being reflected in the title. the included studies in review does not indicate the review is exclusively based on Sydney and Ras Al-Khaimah.
Comments on the Quality of English LanguageNone
Author Response
Comment: The paper has been well-revised but again needs to be more concise and focused on its topic. It remains unclear from the manuscript why two sites in the title, Sydney and Ras Al-Khaimah, which are 12,000 km apart and lack any cultural, geographical, or ethnic similarities, were chosen. A detailed explanation of the rationale behind selecting these diverse locations would enhance the analysis. While this choice might be understandable for primary research, for a review, it could lead to scattered findings due to differences in socio-cultural, medical, economic, and geopolitical contexts.
Response: Thank you for your suggestion. I have elaborated on the above points. Please see Page 2, lines 78-92 for the added changes.
Reviewer 2 Report (Previous Reviewer 2)
Comments and Suggestions for Authors
The authors have addressed the reviewers' comments, however, there are some parts in the text that need to be addressed:
1) Line 149 established criteria, and the context of each study PLEASE DEFINE THE CRITERIA
2) Lines 157-169 should be inserted in Line 138 before the sentence “The final number of studies included…”
PLEASE CHECK THE METHODOLOGY SECTION SO THAT THE TEXT FLOWS IN A LOGICAL ORDER.
3) Lines 426-427 PLEASE REPHRASE AS FOLLOWS It is essential that this approach incorporates other preventive activities such as breast cancer screening, biochemical testing, and blood pressure monitoring.
Comments on the Quality of English LanguagePLEASE CLOSELY CHECK THE GRAMMAR IN THE NEWLY WRITTEN SECTIONS, for example:
Lines 428-429 These interventions will not only help prevent cervical cancer but WILL also have an added value towards general health prevention.
Author Response
The authors have addressed the reviewers' comments, however, there are some parts in the text that need to be addressed:
1) Line 149 established criteria, and the context of each study PLEASE DEFINE THE CRITERIA
Response: Thank you for pointing this out. I have elaborated on the criteria used to select the articles for our literature review. Please see page 4, lines 176-194.
2) Lines 157-169 should be inserted in Line 138 before the sentence “The final number of studies included…”
PLEASE CHECK THE METHODOLOGY SECTION SO THAT THE TEXT FLOWS IN A LOGICAL ORDER.
Response: Thank you for your suggestion. I have moved this section before the sentence on line 138.
3) Lines 426-427 PLEASE REPHRASE AS FOLLOWS It is essential that this approach incorporates other preventive activities such as breast cancer screening, biochemical testing, and blood pressure monitoring.
Response: I have rephrased lines 426-427 as suggested. Please see page 25, lines 494- 496.
PLEASE CLOSELY CHECK THE GRAMMAR IN THE NEWLY WRITTEN SECTIONS, for example:
Lines 428-429 These interventions will not only help prevent cervical cancer but WILL also have an added value towards general health prevention.
Response: I have made this change as suggested. Please see page 25, lines 508-509.
We have proofread all of the newly written sections again to ensure that there are no other grammatical errors.
This manuscript is a resubmission of an earlier submission. The following is a list of the peer review reports and author responses from that submission.
Round 1
Reviewer 1 Report
Comments and Suggestions for Authors
The paper is well-written but need to be more concise and focused on topic.
One of the main concerns is the need for the paper to be more concise and tightly focused on the topic at hand. The narrative, while informative, tends to diverge at points, which could potentially dilute the impact of the key findings and discussions. The manuscript could benefit from a detailed explanation of why Sydney and Ras Al-Khaimah were chosen as the study locations. Understanding the rationale behind selecting these socio-culturally, medically, economically, and geopolitically diverse locations would add depth to the analysis. Are there specific similarities in characteristics or challenges in these locations that make them particularly relevant for this study? If this were primary research, I could understand the setting of collaboration, but given that this is a review, the findings may be very scattered due to differences in the socio-cultural, medical, economic, and geopolitical contexts of these diverse locations. What is the rational behind chose to focus on Sydney and Ras Al-Khaimah, 12000 km apart specifically?
Make a stronger case for the importance of understanding cultural differences in health practices across this area.
More explicitly define the research gaps and the study's specific objectives, particularly focusing on the unique challenges faced by women in these areas, as the population demographic in both areas are diverse.
A deeper focus on the cultural and social factors influencing cervical cancer awareness and screening in these regions would be beneficial.
Clarify any criteria used for including or excluding specific studies.
Describe the search strategy in more detail, including specific keywords and any Boolean operators used, add this as a supplementary material if required.
Elaborate on the process of screening and selecting articles. Include information on how many articles were initially found, how many were screened, and the final number included in the review. If there was a process for resolving disagreements in article selection, describe this process.
Explain how data was extracted from the selected articles and how it was synthesized. Detail the approach used for analyzing and combining findings from different studies.
Mention if and how the quality of the included studies was assessed. This is important for understanding the reliability and validity of the findings drawn from these studies.
Organize the findings into clear sub-sections, such as barriers by category (e.g., cultural, logistical).
Discuss how the identified barriers and facilitators compare to findings in other regions or populations. Interpret the potential reasons behind unique findings in the UAE and Sydney contexts.
Elaborate on how the findings could influence healthcare policy, especially concerning migrant women in Sydney and RAK, given healthcare system are contrastingly different across this two place.
Comments on the Quality of English LanguageNone
Reviewer 2 Report
Comments and Suggestions for Authors
This is a review on the barriers and facilitators migrant women face in their participation in cervical screening activities.
In the introduction the main risk factors for cervical cancer should be mentioned. Some data on the effectiveness of HPV vaccination as well as of cervical cancer screening practices in preventing cervical cancer should be presented.
When referring to the decrease of cervical cancer incidence and mortality in Australia, some information on the proportion of migrant women participating in screening activities would be useful. In lines 76/77 the authors refer to “low” participation. Please give some more information to support this statement.
Lines 118-120 please check the number of the studies, something is wrong: “186 articles were identified, 107 were excluded, one reviewer screened 79 articles, and 245 completed the inclusion criteria”.
Line 125: "The gathered data from each publication was carefully analysed and tabulated in Table 3". Please expand on the way the data was gathered and analysed.
It is important to highlight that cervical cancer screening should be viewed as part of a holistic prevention strategy approach where other preventive activities should be incorporated e.g. breast cancer screening, biochemical testing such blood glucose levels, blood pressure and that the interventions will have an added value towards health prevention in general and not only cervical cancer. Maybe some additional references on the barriers and facilitators of participation of migrant women in such activities should be introduced in order to facilitate the design of effective onterventions.
Line 307 availability of interpreter is debatable in the case of cervical cancer, when we consider discussing sensitive matters. A comment should be made on this.
Line 339 "Unlike a systematic review, a narrative review is not a comprehensive review of the extant literature". What is extant literature?
English editing is necessary as some phrases and grammar need improvement
Comments on the Quality of English LanguageThis review needs some polishing and English language editing before it can be published.